

# Gait changes in a line of mice artificially selected for longer limbs

Leah M. Sparrow[1], Emily Pellatt[1], Sabrina S. Yu[2], David A. Raichlen[3], Herman Pontzer[4] and Campbell Rolian[1,5]

[1] Department of Comparative Biology and Experimental Medicine, Faculty of Veterinary Medicine, University of Calgary, Calgary, Alberta, Canada
[2] Cumming School of Medicine, University of Calgary, Calgary, Alberta, Canada
[3] School of Anthropology, University of Arizona, Tucson, AZ, United States
[4] Department of Anthropology, City University of New York, Hunter College, New York, NY, United States
[5] McCaig Institute for Bone and Joint Health, University of Calgary, Calgary, Alberta, Canada

Corresponding author
Campbell Rolian,
cprolian@ucalgary.ca

## ABSTRACT

In legged terrestrial locomotion, the duration of stance phase, i.e., when limbs are in contact with the substrate, is positively correlated with limb length, and negatively correlated with the metabolic cost of transport. These relationships are well documented at the interspecific level, across a broad range of body sizes and travel speeds. However, such relationships are harder to evaluate within species (i.e., where natural selection operates), largely for practical reasons, including low population variance in limb length, and the presence of confounding factors such as body mass, or training. Here, we compared spatiotemporal kinematics of gait in Longshanks, a long-legged mouse line created through artificial selection, and in random-bred, mass-matched Control mice raised under identical conditions. We used a gait treadmill to test the hypothesis that Longshanks have longer stance phases and stride lengths, and decreased stride frequencies in both fore- and hind limbs, compared with Controls. Our results indicate that gait differs significantly between the two groups. Specifically, and as hypothesized, stance duration and stride length are 8–10% greater in Longshanks, while stride frequency is 8% lower than in Controls. However, there was no difference in the touch-down timing and sequence of the paws between the two lines. Taken together, these data suggest that, for a given speed, Longshanks mice take significantly fewer, longer steps to cover the same distance or running time compared to Controls, with important implications for other measures of variation among individuals in whole-organism performance, such as the metabolic cost of transport.

## INTRODUCTION

In terrestrial legged locomotion, a stride is divided into two phases: stance, when the foot/paw is in contact with the ground, and swing, when the foot leaves the substrate as the limb transitions to the next stance phase. Stance duration is influenced by a number of anatomical and biomechanical factors, including limb length. Limb length is positively correlated with step length, the distance traveled while a foot is in contact with the ground.

As stance duration is simply step length divided by travel speed, for a given travel speed, organisms with relatively longer limbs typically have longer stance durations (*Hoyt, Wickler & Cogger, 2000*). Stance duration is itself an important determinant of whole-organism performance, especially in relation to the metabolic cost of moving the body. During level walking and running, muscle forces produced by the limbs, integrated over the stance phase, must equal bodyweight integrated over the whole stride. Longer stance phases reduce muscle force production rates per unit of body mass, in turn reducing the overall metabolic cost of supporting the body during locomotion (*Kram & Taylor, 1990*; *Roberts et al., 1998*; *Pontzer, 2007a*; *Pontzer, 2016*). Hence, it follows that organisms with longer limbs and stance durations also tend to have a relatively lower mass-specific metabolic cost per unit of distance traveled, often called the cost of transport or COT (J kg$^{-1}$ m$^{-1}$) (*Kram & Taylor, 1990*; *Pontzer, 2007a*).

The relationships between limb length, stance duration (also known as "contact time") and COT are well documented at the interspecific level, e.g., among terrestrial quadrupedal mammals in which limb length and COT data were sampled from the same individuals (*Hoyt, Wickler & Cogger, 2000*; *Kram & Taylor, 1990*; *Roberts et al., 1998*; *Pontzer, 2007a*; *Taylor, Heglund & Maloiy, 1982*; *Taylor et al., 1980*; *Pontzer, 2005*). In contrast, these relationships have been more equivocal at the population level, where differences among individuals in whole organism performance due to variation in these factors have the potential to lead to differential reproductive success (i.e., where natural selection operates (*Arnold, 1983*)). Several studies in human populations have shown a correlation between limb length and gait variables such as stance duration (*Stolze et al., 1997*; *Lythgo, Wilson & Galea, 2009*; *Chapman et al., 2012*), as well as between limb length and COT in walking and running, after controlling for other factors that influence COT, such as body mass (e.g., Refs *Pontzer, 2007b*; *Cavanagh & Kram, 1989*; *Steudel-Numbers & Tilkens, 2004*; *Cavanagh & Kram, 1990*; *Brisswalter, Legros & Durand, 1996*; *Kramer & Sarton-Miller, 2008*, but see also Ref. *Minetti et al., 2000*). Fewer intraspecific studies have examined the relationships between hind limb length, kinematics, and COT within quadrupedal species, and these are limited to comparisons among breeds of dogs or horses. Longer limbs are associated with greater stance duration in dogs (*Voss et al., 2010*; *Voss et al., 2011*) and horses (*Back et al., 1999*; *Galisteo et al., 2001*), but while hind limb length can be predictive of COT in dogs (*Pontzer, 2007b*), taller horses do not necessarily have lower COT (*Griffin et al., 2004*).

One of the practical challenges in relating limb length to locomotor mechanics among individuals *within* populations, is that only a limited range of limb lengths can realistically be sampled, meaning that either variation in skeletal anatomy is too small to detect subtle effects of leg length on gait (i.e., a low signal-to-noise ratio), or impractically large samples are necessary to do so.

Another important limitation of the comparative approach to studying the relationship between limb anatomy and locomotor mechanics—both within and between species—is the complexity of teasing apart the relative impact on gait of body mass, skeletal anatomy, posture and other biological factors (e.g., proportions of muscle fiber types *Zierath & Hawley, 2004*). Within species, many of these factors are genetically and phenotypically correlated, and interact with each other, and with other factors such as age, body
composition (e.g., lean mass), and training, which makes it harder to quantify the relative contributions of limb length alone to gait variation among individuals. For example, among artificially bred dog and horse species, differences in size and limb length are typically correlated with other changes in body proportions and conformation that can affect whole-body kinematics and may thus influence metabolic cost (*Voss et al., 2011*; *Back et al., 1999*). Similarly, among species, fundamental differences in posture (e.g., quadrupedalism vs bipedalism, limb joint extension angles, limb design (e.g., plantigrady vs unguligrady)) and orders of magnitude of differences in body mass may also obscure the effects of limb skeletal anatomy on locomotor mechanics (*Biewener, 1983*).

Here, we use an experimental evolution model to explore the relationship between limb length and locomotor kinematics. Longshanks mice have been selectively bred for increases in tibia length independently of body mass (*Marchini et al., 2014*; *Cosman, Sparrow & Rolian, 2016*). After 14 generations, Longshanks tibiae were on average 14% longer than a random-bred wild-type cohort of the same genetic background (hereafter Controls), but average body masses were the same in both groups. By pooling Longshanks and Control mice, we have created a new population of mice in which the range of variation in limb length is artificially increased, thereby circumventing sample size and resolution issues. More importantly, in this sample the potentially confounding effects of body mass, diet, training, somatic growth, age and even genetic background, are more rigorously controlled. We used the unique Longshanks mouse to investigate the proximate relationships between limb morphology and locomotor mechanics within species. We test the hypothesis that, at a given speed, increased limb length produces predictable changes in gait parameters. Specifically, we predict an increase in stance duration and stride length, and a parallel decrease in stride frequency.

## METHODS

### Samples

All animal procedures were approved by the Health Sciences Animal Care Committee at the University of Calgary (protocol AC13-0077), and were conducted in accordance with best practices outlined by the Canadian Council on Animal Care. We used a sex-balanced sample of adult mice ($156 \pm 17$ days, mean $\pm$ SD) selected at random from 12 Longshanks families (hereafter LS, $n = 11$ females, 11 males) and 12 Control families (hereafter C, $n = 12$ females, 11 males) lines at generation F14. Details of the selective breeding protocol are given elsewhere (*Marchini et al., 2014*). Mice were housed individually in a separate room but in similar conditions to the main colonies of the selective breeding experiment ($22$–$25°$, 50–70% humidity, 12-hour light/dark cycle). Mice were given food and water ad libitum.

### Gait data collection and analysis
#### Gait treadmill procedure
*Training phase:* Beginning at 10 weeks old, each mouse completed a total of eight training trials on a closed chamber treadmill (Columbus Instruments, Columbus, OH) over the course of five months. In the first two trials, the treadmill was inactive, as these trials

allowed the mice to acclimate to the novel treadmill environment. For the remaining six trials, mice completed an initial 5–10-minute habituation period, followed by a running schedule as follows: four minutes at 5 m/min, three minutes at 10 m/min, three minutes at 15 m/min, three minutes at 20 m/min, and a 90 s cool-down period.

Within three days of completing the final training trial, gait data for each mouse were collected using a TreadScan gait treadmill (Cleversys, Reston, VA, USA). Mice were weighed to the nearest 0.01 g prior to being placed in the treadmill. The TreadScan apparatus comprises a transparent, variable-speed treadmill belt (190 mm long, 38 mm wide) enclosed within a Plexiglas chamber. While mice run on the treadmill, their paw placements are recorded on the ventral surface through the use of a mirror placed at 45 degrees to a high-speed camera recording at 100 frames per second. To improve digital tracking of footprints, the contrast between paws and fur was enhanced by applying red food coloring to the plantar surfaces of the paws. The mice were run for approximately three minutes during which three 20-second digital video recordings were taken at each of three test speeds: 10, 15, and 20 m/min. After the first video was captured, each foot was manually traced for approximately 20 frames (equivalent to a full locomotor cycle) to create a foot model for all subsequent analyses.

### Gait variables and analysis

The TreadScan software automatically tracks stance and swing phases of each paw and subsequently generates multiple gait variables from the video footage. A Matlab script (v.2012b; Reston, MA, USA) was first used to extract gait variables from the TreadScan output spreadsheet from all individual strides recorded for each mouse at each speed. For consistency, and because lighting and visibility of kinematics were better for the animal's right side, only gait variables for the right forelimb and hind limb paws were used for analysis. The touch-down sequence was recorded for all four paws (see below).

The following variables were obtained for the right fore- and hind limb paws from the TreadScan output spreadsheets: (1) Stance Time: Time (ms) during which that foot was in contact with the belt; (2) Swing Time (ms): time during which that foot is not in contact with the treadmill, (3) Stride Length (mm): distance travelled between two stance phases of the same foot (As the mice are stationary relative to the treadmill, stride length is estimated as the product of stride time (i.e., stance time plus swing time) and treadmill belt speed (4)), and Stride Frequency (Hz): 1/mean stride duration. To determine whether the increased limb length of Longshanks resulted in a change in its gait sequence relative to Controls, we also obtained timing data for each paw during a full stride cycle. Specifically, using the initiation of stance in the right forepaw as a reference, we obtained the relative timing of touch-down for the other paws, expressed as a percentage of a full stride cycle.

A second Matlab (v.2012b; Reston, MA, USA) script was used to obtain means for each gait parameter in each individual's right fore- and hind limb paws, based on one or more sets of at least three consecutive steps. Outliers, defined as data points that were greater than two standard deviations away from their respective means, were removed and the mean was recalculated. Individuals' gait variable means were based on a minimum of three steps in each limb type and speed condition (mean = 17 steps, range 3–63). Videos of the

slowest speed, 10 m/min, were excluded from our analyses due to the difficulty in selecting a section of video with enough consecutive strides to obtain reliable gait patterns.

Statistical analyses of the gait and gait sequence variables were performed using four separate sets of generalized linear models (GLM). In the first two sets, GLMs were used to test for mean differences between lines (LS1 vs C) and speeds (20 and 25 m/min) in each gait variable *within* a limb type. In the third set of GLMs, we compared mean fore- and hind limb gait variables between the mouse lines at the fastest speed. Finally, we used a GLM to test for mean differences between lines and speeds in gait sequence variables. In all GLMs, we first used a full factorial model, in which line, speed or limb type were treated as categorical predictors, and body mass was included as a continuous predictor (covariate). The models included interaction terms for the respective categorical factors (i.e., Line x Speed, or Line x Limb Type), as well as covariate-by-factor interaction terms (i.e., homogeneity of slopes tests). In all but one variable (see below), none of the covariate-by-factor interaction terms were significant, i.e., the effect of the categorical factors (e.g., speed) on the responses (gait variables) was not dependent on the magnitude of body mass. As argued by Engqvist (*Engqvist, 2005*), in the absence of significant covariate-by-factor interaction terms, GLM analyses should be re-run without these terms, as failing to exclude them implies that the main effects of the factors cannot be generalized over the range of the covariate. Conversely, however, a significant covariate-by-factor interaction term indicates that any significant mean differences among the factors are only true at the intercept, i.e., where the covariate—body mass in this case—is equal to 0.

In the fourth set of GLM analyses, we found a significant interaction between mouse line and body mass in the timing of touch-down of the left hind paw relative to the right forepaw ($F(7, 82) = 4.04$, $p = 0.047$). However, the inclusion of this interaction term had only a small effect on the least squares (LS) means among groups. For example, at the covariate mean (42.93 g), the difference in LS means when the interaction term was excluded vs. included was less than 1% across all groups (range: 0.4–0.9%), while at the 10th and 90th percentile of body mass this difference was less than 10% (range: 4.5–9.5%). Hence, given its modest impact on the actual gait variable response over the covariate range, and to be consistent with the remaining analyses, we excluded all covariate x factor interactions from this GLM. In all GLM analyses, relevant post-hoc pairwise comparisons on all gait variables were made using Tukey's HSD tests. All analyses were carried out in Statistica software (v.10.0, Statsoft, Tulsa, OK).

## Morphometric data collection and analysis

After completing the gait trials, mice were euthanized by $CO_2$ inhalation, weighed and immediately frozen at $-20C$. As part of an ongoing digital tomography archive of the Longshanks experiment, full body micro-CT scans were made using a Skyscan 1173 µCT scanner at a resolution of 45 µm (65–70 kV, 75–105 µA). One of the Longshanks bodies was not recovered following euthanasia. 3D isosurfaces of the scans were produced using Amira v.5.4.2 (Visage Imaging, Berlin, Germany). Bone lengths were determined by calculating the linear distance between 3D digital landmarks placed on specific anatomical features of the limb long bones. Micro-CT measurements are superior to linear measurements from
dissected limbs, as soft tissues are not visible on the µCT scans, and the placement of digital landmarks on homologous anatomical features across individuals is highly repeatable (*Cosman, Sparrow & Rolian, 2016*).

The following anatomical features were used for landmark placements: (1) humerus length—from the center of the proximal articular surface to the distal-most point on the medial epicondyle, (2) ulna length—from the tip of the olecranon process to the tip of the styloid process, (3) carpo-metacarpus—from the proximal articular facet of the central carpal to the tip of the distal articular facet of the third metacarpal, (4) scapula length—from the caudal end of the spine of the scapula to the tip of the center of the glenoid (5) third manual proximal phalanx—from the center of the proximal to the center of the distal articular facets, (6) femur length—from the center of the medial condyle to the tip of the greater trochanter, (7) tibia length—from the anterior-most lip on the proximal epiphysis to the most distal point on the medial malleolus, (8) tarso-metatarsus—from the proximal dorsal border of the centrale tarsal bone to the dorsal border of the distal articular facet of the third metatarsal, (9) third pedal proximal phalanx—from the center of the proximal to the center of the distal articular facets. Limb bone lengths were also summed within limb to obtain forelimb and hind limb anatomical lengths. Two-tailed t-tests were used to compare mean body mass and limb bone lengths between the groups.

## RESULTS

### Morphometric differences

Mean body mass was not significantly different between Control and Longshanks mice in either the test trials or the *ex vivo* CT scanning (Table 1). In the forelimb, all bones were significantly longer in Longshanks when compared to Control mice. Longshanks scapulae were on average 6.6% longer, humeri were ∼12.5% longer, ulnae ∼10.7%, and the hand bones (carpo-metacarpus and third proximal phalanx) were ∼4% longer. When summed across the elements, the anatomical length of the forelimb in Longshanks, including the scapula (*Lilje, Tardieu & Fischer, 2003*; *Schmidt, 2008*) was on average 9.1%, or approximately 4.2 mm, longer than Control forelimbs. Hind limb bones were all significantly longer in Longshanks mice compared to the Control mice: Longshanks mice had a 7.4% longer femur, 14.3% longer tibia (Fig. 1), 9.5% longer tarso-metatarsus, and 9.3% longer pedal proximal phalanx (Table 1). When summed across elements (femur + tibia + foot elements), the Longshanks hind limb was on average 10.7%, or 5.2 mm, longer than Control hind limbs. Selection for relative tibia length in Longshanks thus caused disproportionate changes in the other limb bones, such that overall the Longshanks forelimb increased in length slightly less than the hind limb, although the mean difference in length between the limbs is not significantly different between Longshanks (mean ± SD = 2.84 ± 2.85 mm) and Control (1.89 ± 2.08 mm) (Table 1, $t$-test, $t(42) = 1.27$, $p = 0.21$).

### Gait differences

*Line and speed effects on gait within limbs*: The first two sets of GLMs indicate that body mass was not significantly correlated with any gait variable in the forelimb, but in the hind limb was significantly negatively correlated with swing duration, and positively correlated

**Table 1  Morphometric data: body masses at the gait (Treadscan) and μCT scanning stages, and fore- and hind limb bone lengths between Control and Longshanks mice, expressed as means (SEM).** Significance of the difference in means for all variables was determined using two-tailed *t*-tests. One Longshanks body was not recovered from euthanasia prior to scanning ($n = 21$).

|  | Longshanks ($n = 21$) | Control ($n = 23$) | Statistic |
|---|---|---|---|
| Body mass (gait trials) (g) | 43.73 (1.31) ($n = 22$) | 42.18 (1.27) | $t = 0.84, df = 43, p = 0.40$ |
| Body mass (μCT scan) (g) | 45.81 (1.43) | 44.55 (1.60) | $t = 0.58, df = 42, p = 0.56$ |
| Scapula | 13.83 (0.15) | 12.97 (0.08) | $t = 5.07, df = 42, p < 0.001$ |
| Humerus | 13.65 (0.11) | 12.17 (0.07) | $t = 12.02, df = 42, p < 0.001$ |
| Ulna | 17.02 (0.13) | 15.36 (0.08) | $t = 11.34, df = 42, p < 0.001$ |
| Carpo-metacarpus | 4.43 (0.04) | 4.26 (0.03) | $t = 3.69, df = 42, p < 0.001$ |
| Manual proximal phalanx 3 | 2.14 (0.02) | 2.06 (0.02) | $t = 2.89, df = 42, p < 0.01$ |
| Femur | 18.5 (0.25) | 17.2 (0.17) | $t = 4.39, df = 42, p < 0.001$ |
| Tibia | 21.44 (0.33) | 18.75 (0.18) | $t = 7.37, df = 42, p < 0.001$ |
| Tarso-metatarsus | 10.62 (0.17) | 9.7 (0.10) | $t = 4.79, df = 42, p < 0.001$ |
| Pedal proximal phalanx 3 | 3.35 (0.06) | 3.06 (0.03) | $t = 4.47, df = 42, p < 0.001$ |
| Forelimb | 51.07 (0.38) | 46.82 (0.20) | $t = 10.11, df = 42, p < 0.001$ |
| Hind limb | 53.91 (0.76) | 48.71 (0.43) | $t = 6.11, df = 42, p < 0.001$ |

**Table 2  Standardized coefficients (betas, with standard errors) for each predictor variable in the linear model for each gait variable in each limb.** For Line, factor levels are Control = 0, Longshanks = 1, for Speed, factor levels are 15 m/mi $n = 0$, 20 m/mi $n = 1$. Covariate-by-factor interaction terms were excluded from the analyses (see methods). The gait sequence data are shown as a fraction of the full stride cycle for the right forepaw.

|  | Effect | Swing (ms) | Stance (ms) | Stride length (mm) | Stride freq. (1/s) |
|---|---|---|---|---|---|
| **FORELIMB** | Mass | 0.102 (0.098) | 0.006 (0.064) | 0.068 (0.085) | −0.057 (0.068) |
|  | Line | 0.183 (0.098) | 0.296 (0.064)** | 0.362 (0.085)** | −0.293 (0.068)** |
|  | Speed | −0.373 (0.098)** | −0.752 (0.063)** | 0.503 (0.084)** | 0.720 (0.067)** |
|  | Line*Speed | 0.051 (0.098) | −0.082 (0.063) | 0.010 (0.084) | −0.034 (0.067) |
| **HIND LIMB** | Mass | −0.308 (0.104)* | 0.2 (0.065)* | 0.023 (0.084) | −0.030 (0.064) |
|  | Line | 0.033 (0.104) | 0.27 (0.065)** | 0.358 (0.084)** | −0.312 (0.064)** |
|  | Speed | −0.021 (0.103) | −0.72 (0.065)** | 0.522 (0.083)** | 0.745 (0.064)** |
|  | Line*Speed | −0.06 (0.103) | 0.006 (0.065) | 0.015 (0.083) | −0.002 (0.064) |

**Notes.**
*$p < 0.05$.
**$p < 0.001$.

with stance duration (Table 2). Both speed and line type had significant effects on forelimb and hind limb stance duration, stride length and stride frequency (Table 2, all standardized slopes significantly different from zero, $p < 0.001$). In the forelimb, speed had a significant effect on swing duration, while in the hind limb there was no change in swing duration due to speed (Table 2). There was no significant interaction between line type and speed in either limb, indicating that running faster did not affect gait variables in Longshanks and Control mice differently ($0.19 < p < 0.92$).

Pairwise comparisons between lines indicate that at the lower speed, mean stance duration was significantly longer, by 10.7% in the forelimb and 9.1% in the hind limb in Longshanks compared to Controls (Table 3 and Fig. 2). At the faster speed, the difference

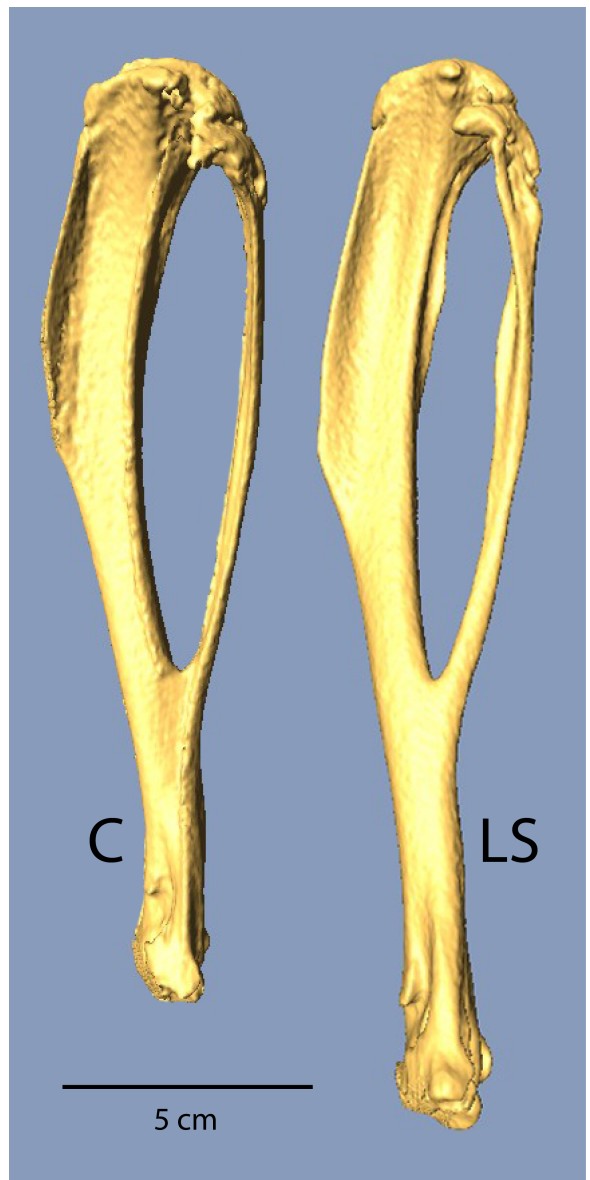

**Figure 1** **μCT scans of Longshanks and Control tibiae.** μCT scans of two individuals from the study sample closest to mean raw tibia length in Control (C, 18.85 mm) and Longshanks (LS, 21.45 mm).

between lines in mean forelimb stance duration (7.5%) trended towards significance (Tukey's HSD, $p = 0.087$), while in Longshanks, hind limb stance duration was over 12.7% longer (Tukey's HSD, $p < 0.001$). Swing durations did not differ between the lines in either limb, however forelimb swing times were significantly shorter at the faster speed (Table 3 and Fig. 2). Mean forelimb and hind limb stride lengths are greater by 7–8% in Longshanks at both speeds, while mean stride frequency in both limbs at both speeds is decreased by 7–8% in Longshanks.

*Line and limb effects on gait within speed:* The GLM analysis with limb type and line as factors (fast speed only) revealed a significant effect of limb type on swing and stance

**Table 3  Limb gait parameters at 15 and 20 m/min in Longshanks (LS, N = 22) vs Control (C, N = 23) mice.** Data reported as least squares means ± SEM, based on a full factorial linear model, with Line and Speed as categorical factors, and body mass as a continuous predictor (see Table 2). Significant differences in pairwise comparisons of means were determined using post-hoc Tukey's HSD tests. Statistical significance ($p < 0.05$) of mean differences between lines within speed are indicated in bold, and between speeds within line with an asterisk. ($p < 0.05$).

| | Speed | Line | Swing (ms) | Stance (ms) | Stride length (mm) | Stride freq. (1/s) |
|---|---|---|---|---|---|---|
| FORELIMB | 15 m/min | C | 94.07 (2.66)* | **147.91 (2.64)*** | **60.50 (1.15)*** | **4.15 (0.07)*** |
| | | LS | 97.67 (2.72)* | **163.8 (2.7)*** | **65.35 (1.18)*** | **3.87 (0.07)*** |
| | 20 m/min | C | 82.39 (2.66)* | 119.77 (2.64)* | **67.31 (1.15)*** | **4.98 (0.07)*** |
| | | LS | 88.83 (2.72)* | 128.75 (2.7)* | **72.45 (1.18)*** | **4.61 (0.07)*** |
| HIND LIMB | 15 m/min | C | 77.07 (3.48) | **166.42 (3.67)*** | **60.87 (1.11)*** | **4.12 (0.06)*** |
| | | LS | 80.25 (3.56) | **181.57 (3.76)*** | **65.44 (1.13)*** | **3.78 (0.07)*** |
| | 20 m/min | C | 78.4 (3.48) | 124.73 (3.67)* | **67.63 (1.11)*** | **4.95 (0.06)*** |
| | | LS | 77.5 (3.56) | **140.55 (3.76)*** | **72.61 (1.13)*** | **4.60 (0.07)*** |

**Table 4  Comparison of fore- and hind limb gait parameters at 20 m/min in Longshanks (LS, N = 22) vs Control (C, N = 23) mice.** Data reported as least squares means ± SEM, based on a full factorial linear model, with Line and Limb Type as categorical factors, and body mass as a continuous predictor. Significant differences in pairwise comparisons of means were determined using post-hoc Tukey's HSD tests. Statistical significance ($p < 0.05$) of mean differences between lines within limb type are indicated in bold, and between limb types within line with an asterisk. ($p < 0.05$).

| Limb type | Line | Swing (ms) | Stance (ms) | Stride (ms) | Stride length (mm) | Stride freq. (1/s) |
|---|---|---|---|---|---|---|
| FORE | C | 82.04 (3) | 120.12 (2.76) | **202.16 (3.32)** | 67.31 (1.11) | 4.97 (0.07) |
| | LS | 89.2 (3.07)* | 128.39 (2.82)* | **217.59 (3.39)** | 72.45 (1.13) | 4.61 (0.07) |
| HIND | C | 78.88 (3) | **124.37 (2.76)** | 203.25 (3.32) | 67.68 (1.11) | 4.95 (0.07) |
| | LS | 77 (3.07)* | **140.93 (2.82)*** | 217.92 (3.39) | 72.56 (1.13) | 4.61 (0.07) |

durations in the Longshanks mouse, with the forelimb having a relatively longer swing phase and shorter stance phase compared to the hind limb (Table 4, Tukey's HSD, $p < 0.05$). As expected, however, stride duration, stride length and stride frequencies are the same between the fore- and hind limb within line. In other words, in Longshanks, stance duration as a percentage of stride duration (i.e., duty factor) is different in the forelimb and hind limb, but fore- and hind-limb cycles are of equal duration.

*Line and speed effects on gait sequence:* The mean relative timing of paw touch-downs is shown in Table 5, and gait sequences are shown in Fig. 3. The GLM analysis showed no effect of line on gait sequence data, but speed had a significant effect on the gait sequence of the forepaws (standardized beta $= -0.23$, $F_{4,85} = 4.21$, $p = 0.03$), with contact of the contralateral forepaw occurring 4–5% earlier at the faster speed in both lines (Tukey's post-hoc HSD, $p = $ n.s.). Combining the touch-down sequence and stance duration data at 20 m/min, both lines have very similar gait sequence profiles, although Longshanks mice have relatively longer hind limb stance phases (greater duty factors, Fig. 3).

## DISCUSSION AND CONCLUSION

Gait is influenced by a number of anatomical (e.g., mass, skeletal size) and biomechanical factors (e.g., speed, bipedal vs. quadrupedal locomotion). Limb length is positively correlated with step length, and hence with stance duration, in terrestrial species across a
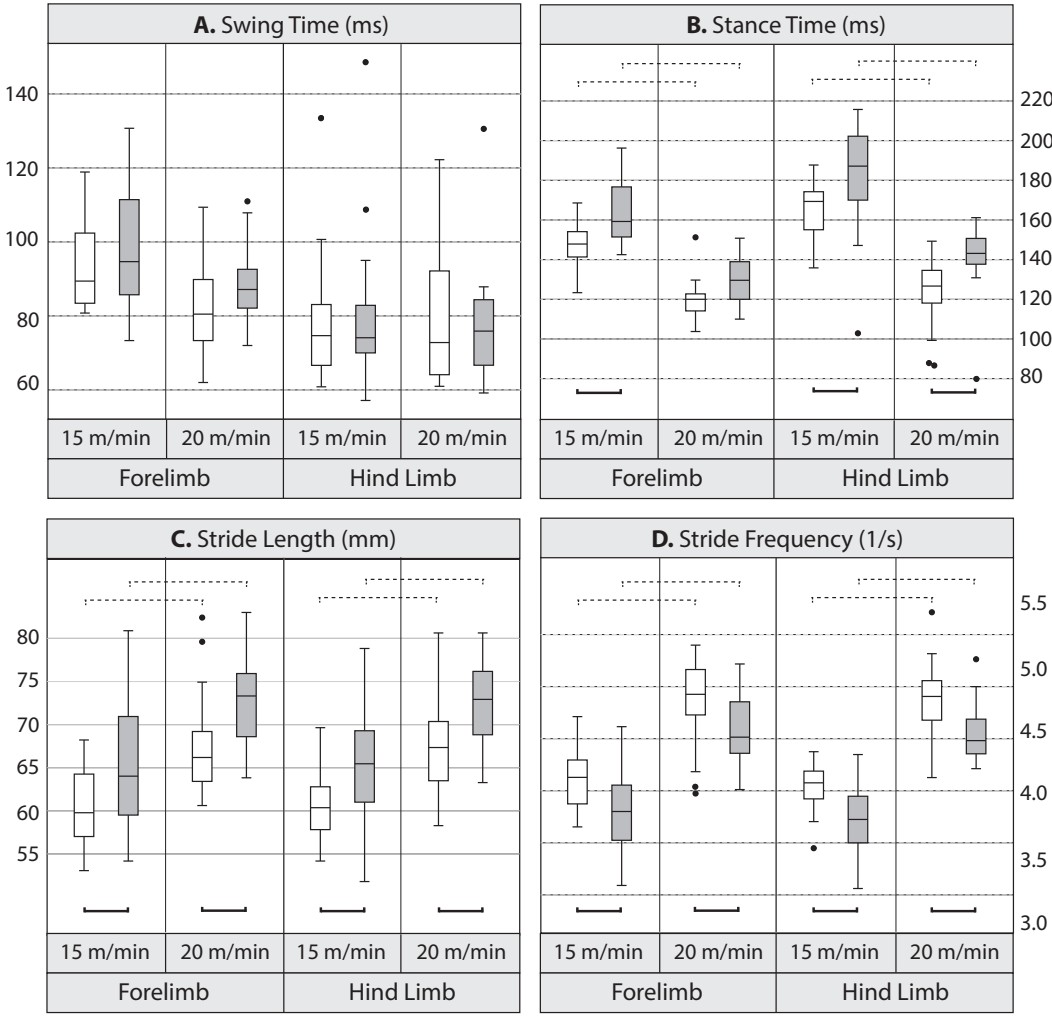

Figure 2 **Boxplots.** Comparison of gait variables in Longshanks and Control mice at 15 and 20 m/min. Boxplots of swing duration (A, in milliseconds), stance duration (B, in milliseconds), stride length (C, in mm), and stride frequency (D, in seconds$^{-1}$) in Longshanks (shaded box) and Control mice (open box). Horizontal lines within boxes represent medians, boxes indicate interquartile ranges, whiskers indicate non-outlier ranges, and outliers are indicated with black dots. Solid brackets below boxplots denote statistically significant mean differences between the lines within a speed, dotted brackets above boxplots indicate statistically significant differences between speeds within lines (at the $p < 0.05$ level). For clarity, differences between limbs within speed are not indicated (see Table 4).

broad range of body sizes and limb lengths (*Hoyt, Wickler & Cogger, 2000*; *Kram & Taylor, 1990*; *Roberts et al., 1998*). Determining if this relationship also holds within populations is difficult, because of the challenge of sampling adequate variation in limb bone lengths that is not also correlated with other variables that influence gait.

We used the long-limbed Longshanks mouse line to test the hypothesis that these mice would have longer strides, longer stance durations and thus lower stride frequencies when compared to mice from a random-bred control group. Crucially, these two groups have been raised under the same conditions and were trained on the treadmill using identical protocols. In addition, they have the same average body mass, and come from the same

**Table 5 Comparison of gait sequences in Longshanks and Control.** Data reported as means ± SEM, based on a full factorial linear model, with Line and Speed as categorical factors, and body mass as a continuous predictor. Means represent the proportion of a full stride cycle of the right forepaw (from 0 = stance initiation to 1 = stance initiation of the next cycle) at which the other paws initiate their stance phases. No significant differences were found between lines within speed, nor between speeds within line (Tukey's HSD tests).

| Speed | Line | Ipsilateral (right) hind paw | Contralateral (left) forepaw | Contralateral (right) hind paw |
|---|---|---|---|---|
| 15 m/min | C | 0.63 (0.01) | 0.53 (0.01) | 0.16 (0.01) |
| | LS | 0.63 (0.01) | 0.54 (0.01) | 0.16 (0.01) |
| 20 m/min | C | 0.63 (0.01) | 0.51 (0.01) | 0.15 (0.01) |
| | LS | 0.63 (0.01) | 0.51 (0.01) | 0.14 (0.01) |

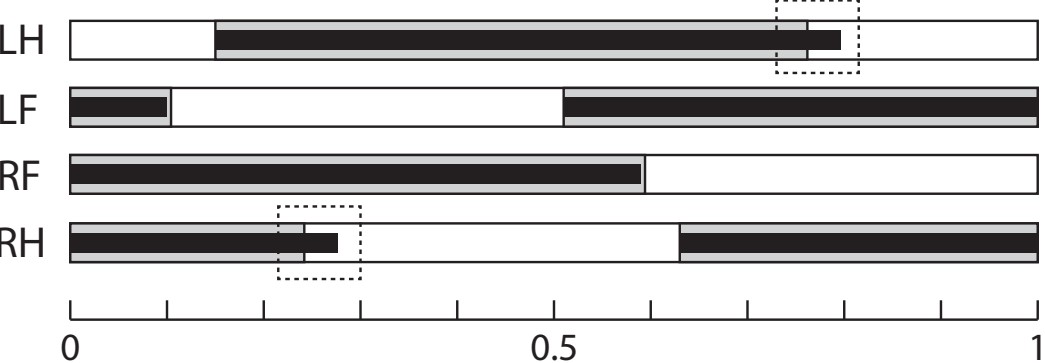

**Figure 3 Gait Sequence.** Mean gait sequence in Control (grey boxes) and Longshanks (thin black boxes). The solid boxes represent the stance phases of each paw, the white boxes the swing phases. Total length, from 0 to 1, represents a full stride cycle of the right fore paw, where 0 touch-down and 1 = touch-down of the following cycle. The stance phase durations of the left paws are duplicated from the right side, as only the latter were analyzed (see methods). Dashed boxes highlight the slightly longer stance time relative to stride time in the hind limbs of Longshanks. Abbreviations: LH, left hind paw; LF, left forepaw; RF, right forepaw; RH, right hind paw.

genetic background. Hence, we were able to isolate the effects of limb bone length on gait parameters from other potentially confounding factors such as training, speed and body mass.

Our results provide strong support for our hypothesis, in both limbs: at the lower speed Longshanks had 9–10% longer stance durations, 7–8% longer strides and stride durations, and thus 7–8% lower stride frequencies. At the faster speed, swing durations in the forelimb were reduced in both lines, while Longshanks' stance duration for the hind limb was longer by 12.7%. Overall, however, the differences in stride length and duration, and stride frequency between the lines in both limbs remained 7–8% at 20 m/min. Interestingly, there was no difference between Longshanks and Control in swing times of either limb, at either speed. Equivalent swing times suggest that, despite its increase in bone lengths, the natural swing period of the limbs in Longshanks have not changed substantially relative to Controls, although this could also be due to greater muscle work to move the limb during the swing phase in Longshanks. Similar natural periods could result from a similar

distribution of muscle mass in both lines, despite the increased limb length in Longshanks. Future work will more carefully assess why the mean swing phase durations do not differ between these populations.

The gait sequence data show that the increase in limb length in Longshanks did not impact the relative timing and sequence of stance initiation in the four limbs at 20 m/min (Table 5 and Fig. 3). Moreover, the forelimb duty factors are very similar in both lines at this speed (~0.59, Table 4). However, the relatively longer stance duration of the Longshanks hind limbs at 20 m/min increases its duty factor slightly relative to Controls (0.647 vs 0.612, Table 4). The reasons for the longer stance duration in the Longshanks hind limb are unclear, but may relate to the fact that overall the Longshanks hind limb increased in length slightly more than the forelimb as a result of selection on tibia length (Table 1).

One limitation of this study is the absence of kinematic data from lateral views of the limbs in these mice, which precludes us from determining precisely if and how limb protraction, retraction, and joint angles have changed in Longshanks as a consequence of changes in the relative lengths of its limb bones (Table 1). As a first approximation, we used our morphometric and gait data to estimate differences in joint angles in the hind limbs of Longshanks vs Controls, based on preliminary limb angular excursion data. Step length is the horizontal distance the hip travels while the paw is in contact with the substrate. On a treadmill, the proximal joints are effectively stationary, and step length represents instead the distance traveled by the paw while in contact with the treadmill belt. Stance on a treadmill can thus be modeled as a "support triangle," described by the hip joint (vertex) and the points of paw-belt contact at touch-down and toe-off (Fig. 4). The base of the triangle represents step length, the height is the vertical distance between the hip and substrate. The triangle's sides represent the hind limb at touch-down and toe-off, and their lengths at these time points are determined by the limb's joint angles and bone lengths.

When the limb angular excursion (i.e., the vertex angle) is known, the lengths of the triangle's sides can be estimated. We obtained hind limb protraction and retraction angles from a small sample of Longshanks and Control mice from generation F22 ($n = 5$ each), running at 20 m/min (Fig. S1). The hind limb protraction, retraction, and excursion angles did not differ significantly between the lines (Table S1), and the overall mean excursion angle was 74.7° ± 2.8 (mean ± SEM). Thus, even though step length in Longshanks hind limbs was 12.7% longer, the angles in the support triangle in both lines were the same, implying that the hind limb support triangle's sides in Longshanks are all also ~12.7% longer. Using the sine rule, we estimated the mean triangle side length at touch-down (i.e., protraction) to be 38.91 mm in Controls, and 43.85 mm in Longshanks, while at toe-off (retraction) mean lengths were 28.09 mm and 31.72 mm, respectively (Fig. S1).

Combining these support lengths with the mean lengths of the femur, tibia and tarso-metatarsus in each line (Table 1), and assuming that the hind paw at touch-down is horizontal and at toe-off the tarso-metatarsus is perpendicular to the treadmill (Fig. S1), we solved graphically for mean knee and ankle joints in Longshanks and Control (Fig. 4). Mean angles in Longshanks hind limbs are all very similar to Controls. The greatest difference is in the knee at touch-down, which is ~5°, or 4%, more extended in Longshanks than in Control mice. This small difference may be due to the fact that the tibia

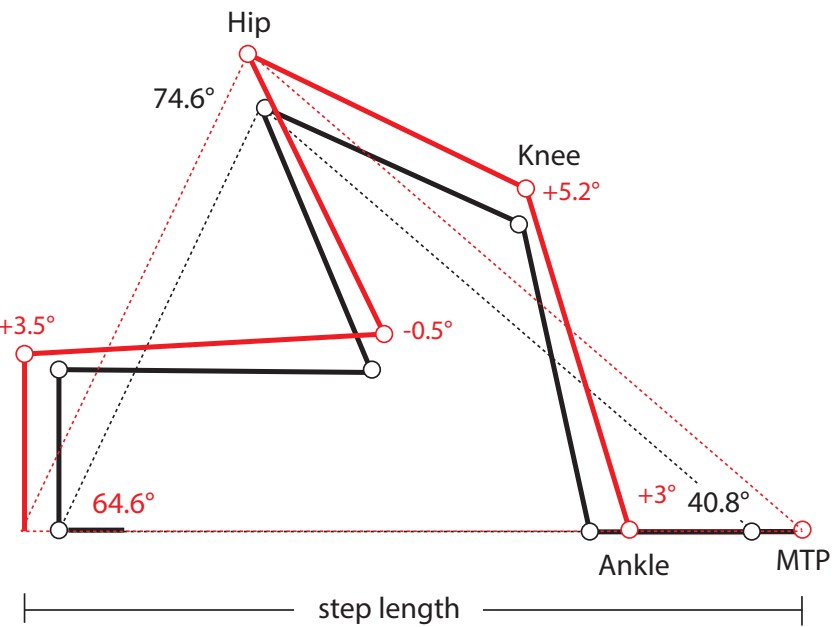

**Figure 4** **Support Triangle.** Diagram of the support triangle in Control (black) and Longshanks (red), obtained using mean step length and the mean protraction, retraction and excursion angles from an unrelated sample of mice from both groups ($n = 5$ each). Step length (base of the support triangle) and these angles were used to obtain the lengths of sides of the support triangle in each group. Using the mean long bone lengths in each group (Table 1), we then solved graphically for angles at the ankle and knee, on the assumption that the tarso-metatarsus was approximately flat at touch-down, and perpendicular to the treadmill at toe-off (Fig. S1).

and femur in Longshanks did not increase in length proportionately (+14.3% vs +7.4%). Overall, the 10.7% increase in hind limb length in Longshanks likely accounts for much of the 12.7% increase in its hind limb step length, and necessitates only minor joint extensions at the knee and ankle in Longshanks (Fig. 4). Limb angular changes in the forelimb are likely equally small, however given that the mean increase in forelimb step length (+7.5%) is less than the overall increase in forelimb length (+9.1%), the joint angular changes in that limb in Longshanks more likely entailed flexion rather than extension, absent any changes in forelimb excursion angles between the lines. More precise limb kinematics will clearly be required to confirm these data and determine the combination of changes adopted by Longshanks to increase step and stride lengths in both limbs.

Our findings have implications with respect to the cost of transport in Longshanks. The derived morphology of the Longshanks mouse altered its gait in a predictable fashion, resulting in increased stance durations and stride lengths relative to Controls. Both of these characteristics contribute to the cost of transport in terrestrial organisms. Increased stance time distributes the muscle forces necessary to support body weight over a longer interval, lowering the *rate* of muscle force production, i.e., the volume of muscle recruited per distance traveled (*Kram & Taylor, 1990*; *Roberts et al., 1998*; *Pontzer, 2007a*; *Pontzer, 2016*). Complementary to this, increasing stride length reduces the number of steps required to cover a given distance, reducing activation costs (*Pontzer, 2007a*). Thus, our gait data

predict that, all else being equal, including limb and joint angles as estimated above, the mass-specific cost of transport in Longshanks should be lower when compared with Control mice. We are currently testing this hypothesis with the use of a metabolic treadmill in Longshanks and Control.

Variation in whole organism performance directly impacts evolutionary fitness, and is an important driver of adaptive evolution (*Arnold, 1983*; *Irschick, 2003*; *Irschick et al., 2008*; *Dalziel, Rogers & Schulte, 2009*). In the process of selecting for a target morphological trait (longer tibiae relative to body mass), we have "forward engineered" a morphology in Longshanks with a quantifiable impact on whole organism *biomechanical* performance. We do not yet know the impact of the observed change in gait on *physiological* performance (e.g., metabolism, especially cost of transport), and it is challenging to relate any differences in these measures of whole organism performance to differences in survival and reproductive success of Longshanks and Control mice in the wild, as these are laboratory-reared animals raised in highly controlled and homogeneous environments. Nevertheless, this study demonstrates the relationship of selectable phenotypic variation in skeletal anatomy to variation among individuals in whole organism performance, in the form of locomotor mechanics. Hence, it provides an important link between population-level, microevolutionary processes and the adaptive origins of macroevolutionary diversity in limb musculoskeletal anatomy among terrestrial mammals.

## ACKNOWLEDGEMENTS

Thanks are due to Jason Anderson and Jessica Theodor for providing access to the SkyScan 1173 µCT scanner, and to Kevin Chapman in the Behavioral Core Facility in the Cumming School of Medicine (University of Calgary) for assistance with the TreadScan system. John Bertram and Douglas Syme provided useful feedback at several stages of the study, and their help is gratefully acknowledged. We also wish to thank three reviewers and the associate editor, for providing feedback and suggestions on a previous version of the manuscript.

### Funding

Leah Sparrow was funded by a Queen Elizabeth II Scholarship from the University of Calgary. Emily Pellatt was supported by an award from the Markin Undergraduate Summer Research Program, Sabrina Yu was funded by the Heritage Youth Researcher Summer (HYRS) program from Alberta Innovates Health Solutions. Campbell Rolian was funded by a Discovery Grant from the Natural Sciences and Engineering Research Council of Canada (NSERC), and by the Faculty of Veterinary Medicine at the University of Calgary. The funders had no role in study design, data collection and analysis, decision to publish, or preparation of the manuscript.

## Grant Disclosures

The following grant information was disclosed by the authors:

University of Calgary.

Markin Undergraduate Summer Research Program.

Heritage Youth Researcher Summer (HYRS).

Discovery Grant from the Natural Sciences and Engineering Research Council of Canada (NSERC).

Faculty of Veterinary Medicine at the University of Calgary.

## Competing Interests

The authors declare there are no competing interests.

## Author Contributions

- Leah M. Sparrow conceived and designed the experiments, performed the experiments, analyzed the data, wrote the paper, prepared figures and/or tables, reviewed drafts of the paper.
- Emily Pellatt and Sabrina S. Yu performed the experiments, reviewed drafts of the paper.
- David A. Raichlen and Herman Pontzer wrote the paper, reviewed drafts of the paper.
- Campbell Rolian conceived and designed the experiments, analyzed the data, wrote the paper, prepared figures and/or tables, reviewed drafts of the paper.

## Animal Ethics

The following information was supplied relating to ethical approvals (i.e., approving body and any reference numbers):

All animal procedures were approved by the Health Sciences Animal Care Committee at the University of Calgary (protocol AC13-0077), and were conducted in accordance with best practices outlined by the Canadian Council on Animal Care.

## Data Availability

The raw data has been supplied as a Supplementary File.

## Supplemental Information

Supplemental information for this article can be found online at http://dx.doi.org/10.7717/peerj.3008#supplemental-information.

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
