# Peer review of "Gait changes in a line of mice artificially selected for longer limbs"

_PeerJ, doi:10.7717/peerj.3008_

## Round 0.1 · original submission · Major Revisions

· Academic Editor

Major Revisions

I have received three reviews that range from Minor to Major Revisions. I disagree with Reviewer 3 who questions the appropriateness of this study for PeerJ, and thus look forward to your revisions in response to the reviewers. I have marked this manuscript as Major Revisions in accordance with the majority opinion, and provide you with the opportunity to revise and respond to the specific and general critiques of the reviewers. I encourage you to take the reviews as an indication of the reception of the paper.

Reviewer 1 provides an number of targeted questions regarding the statistical analysis.

Reviewer 2 has important questions regarding the overall scope of the implications for evolutionary issues and potentially additional studies that may be relevant in the discussion.

Reviewer 2 & 3 have suggestions for additional data or analyses that could have been conducted. Please revise or address why these data were not included and potential implications.

Again, I look forward to receiving you revised manuscript.

Best,

Phil

·

Basic reporting

This is a very well-written, and concise, manuscript. I have no critiques regarding basic reporting.

Experimental design

I have a couple of minor quibbles regarding experimental design. First, as the authors state in the discussion, the chain leading from limb length to stride length, stride frequency, and ultimately COT has more to do with effective limb length (and hence posture) than with morphological limb length per se. I understand that the Treadscan system doesn't provide joint kinematics, and approve of how the authors handled this potential confound in their discussion. However, I think the explanation would be stronger if they provided a figure illustrating their "triangle" argument about the determinants of stride length. I know these kind of figures abound in the literature (I'm thinking of Fig 7 Reynolds's 1987 paper, "Stride length and its determinants in humans, early hominids, primates, and mammals"), but putting one in here nonetheless would make things clearer to the reader.

Second, I was curious why the authors didn't test for covariate-by-factor interactions in their GLM analysis (i.e., line by body mass, or speed by body mass, or even line by speed by body mass). In other words, does the relationship between the gait variables and body mass differ between lines, or between speeds, or between lines at different speeds? If these covariate interactions are significant, it would preclude from performing simple tests of the main effects.

Finally - really minor - how was stride length calculated if the animals were on a treadmill and, presumably, not exhibiting any net forward movement of the center of mass? Was it calculated from belt speed and stride duration? Please elaborate.

Validity of the findings

All of the results appear valid and the interpretations sound. Aside from my minor point about checking for covariate interactions (see above) I see nothing the authors could fix.

Reviewer 2 ·

Basic reporting

The article fails to cite basic key texts on the relationship between lower limb length and locomotion, as it pertains to evolution. In fact, the authors state that within- population studies on differences in limb length have not been successful in teasing out how selection pressures might work, yet multiple papers have been successful, namely work by Patricia Kramer and Karen Steudel-Numbers (who control for mass and vary limb lengths), as well important experimental work by Ted Garland (who also has specific lines of mice with changes in morphology (as well as physiology)). The failure to even cite these papers, and to suggest that important work on infraspecific variation on locomotion hasn't been done is misleading to potential readers of this paper.

The relevance of this experiment seems to be located in the idea that changes in limb length have the potential to change fitness; in fact the authors argue that this is what gives this heavily controlled sample an advantage over other samples of convenience, yet there are no tests here of reproduction or fitness so it seems as if the experiment is incomplete.

Experimental design

It is not clear why the authors failed to weigh the limbs after sacrificing the animals. If the limbs are different lengths and total body mass is the same, doesn’t this mean that the limbs are either different masses or different shapes, which would then suggest that they have different inertial properties? Given the variation in swing phases, knowing more about the shape of the limbs might aid the authors in uncovering what is happening throughout the gait cycle.

It is also not clear why the authors scanned the limbs instead of just dissecting them to measure the bones directly.

Validity of the findings

Given that speed was constant, and stride frequency is speed divided by stride length, why does stride length increase by 10% but stride frequency decrease by 7%?

Additional comments

It doesn't seem as if this part of the larger set of experiments really does what the introduction and conclusion hopes it will. Either the context needs to be narrowed to exactly what you think can be solved with kinematics, or you need to loop this part of the experiment in with the physiology and breeding studies to get at some of the larger issues at play.

Reviewer 3 ·

Basic reporting

Reported beautifully. Well written paper

Experimental design

Great. Perfect use of controls and experimental group. What we all strive for in locomotor studies. Greater range of locomotor variables encouraged.

Validity of the findings

Great. Exactly what I expected from the title.

Additional comments

Gait changes in a line of mice artificially selected for longer limbs
I thank the editor for the opportunity to review for PeerJ. This was such an enjoyable paper to read, and I am glad this work will soon enter the scientific literature. Overall, I have very few comments. I think this work is creative, well-done, and adds a great deal to how we think about morphological variables and locomotor performance. That being said, I found the article to be somewhat uninteresting. The authors only collected the bare minimum of locomotor gait variables, where a more complete suite was certainly possible and I think should be encouraged.

The authors describe only spatiotemporal gait variables, and of them only three were significant between mice stains, and of those three two of those variables (stride duration and stride frequency) are directly related to each other. The authors go out of their way to explain why kinematic variables could not be collected, but as a reviewer I would want the authors to at least try and present this data and then explain to the readers the limitations of the kinematic analysis rather than not providing data at all. The explanation for how these mice deal with these strange proportions would be incredibly valuable to our understanding of the evolution of increased limb length. Perhaps, the inclusion of gait sequence data would be more interesting. Do the presence of Lateral sequence lateral couplet increase with limb length? This would be very interesting. Related to this accommodation question, surely the forelimb of these mice also accommodate in some way, or did the mice behave like sifaka walking quadrupedally (Granatosky et al., 2016)? This would be great if the authors could address this potential mismatch between the forelimbs and the hind limbs.

In the end, I’m not sure what my opinion is of this article. I really have no specific problems. I think it is well done, reads easy, and is solid, but I it also appears that the authors did the bare minimum to get an article published. I believe this article is worthy of publication, but without the inclusion of other gait variables, perhaps a lower impact factor journal may be more appropriate.

---

## Round 0.2 · Minor Revisions

· Academic Editor

Minor Revisions

Thank you for suitably addressing the reviewers' concerns in your revised manuscript. You paper is essentially accepted, however I have attached an annotated pdf with a number of edits and suggestions for typos, grammar and clarity. Note that I edited your manuscript across two computers so that those attributed to "Reviewer" are also mine. Address these and double check the manuscript closely. I look forward to your returned manuscript.

---

## Author Rebuttal · Round 0.2

Campbell Rolian, PhD
University of Calgary
3330 Hospital Dr NW
Calgary, AB, T2N4N1
Canada

January 16 2017
Dr. Philip Reno
Academic Editor, *PeerJ*

**RE: *PeerJ* MS # 12093 – "Gait changes in a line of mice artificially selected for longer limbs"**

Dear Dr. Reno (Phil),

On behalf of my co-authors, I wish to thank the reviewers for their thoughtful comments on our manuscript, and to thank you for your patience while we addressed these comments. As you will see, we have made substantial changes to the manuscript, including new analyses focused on the forelimb, on gait sequence data, as well as on joint angular kinematics. As a result, we believe the manuscript provides a more complete picture of gait changes in the Longshanks mouse. Below you will find an overview of general changes, along with our detailed response to reviewers' suggestions, which we have paraphrased for brevity. Note that, in the text, significant changes or additions have been highlighted in blue font.

## General changes:

Major changes: (1) We extracted gait variables from *all* usable stride cycles in the raw data output from Treadscan for each individual, including forelimb variables that match the original data from the hind limb, as well as data on the sequence of paw touch-downs for all paws. (Lines 125-147). We removed duty factor data, as this did not add information that could not already be obtained from swing and stance durations. (2) We re-ran our statistical analyses to include covariate-by-factor interactions in the generalized linear models (GLM) (see Reviewer 1 comments) (Lines 150-176). (3) We measured forelimb long bone lengths from the CT data, and remeasured hind limb segments to include the tarsal region into a tarso-metatarsus which more accurately reflects the length of the foot. For the same reason, we changed the phalangeal measurement to the third digit, which is the longest segment of the distal forepaw and hind paw (Lines 187-199). Statistical analyses of these new morphometric measurements are presented in an updated Table 1.

(4) Results of the GLM analyses are presented for effects of line and speed on gait within each limb (Lines 219-236, Tables 2-3, new Figure 2), effects of line and limb type on gait at 20 m/min only (Table 4, similar results obtain at 15 m/min) and line and speed effects on gait sequence data relative to the right forepaw (Table 5, new Figure 3). The figure and table captions have been updated to reflect these changes.

(6) In our updated discussion, we now include preliminary joint angle data collected from lateral video footage of a small sample of unrelated mice (n = 5 per group), in order to estimate joint flexion/extension angle differences between Longshanks and Control, based on the mean effective hind limb length derived from the

morphometric measurements (Lines 286-323, new Figures 4 and S1, Table S1). As these data were collected by another student (Emily Pellatt), we have now included her as a co-author on the study.

Minor changes: we noticed a discrepancy between the belt speeds entered on the treadmill controller (15 and 20 m/min), and the speeds used by the TreadScan software for calculations of stride length (20 and 25 m/min). We verified manually that the controller-based speeds are correct, whereas we used the incorrect TreadScan values in our original draft. We have updated these values and adjusted the stride lengths accordingly. None of the results for these analyses are impacted by this adjustment. I have also updated my affiliation to include the McCaig Institute for Bone and Joint Health.

## Reviewer-specific changes:

### Reviewer 1

1) A "support triangle" figure could be included in the discussion to better illustrate the determinants of stride length in relation to functional vs anatomical limb length.

> *Authors' response*: Based on the recommendation of Reviewer 3, who suggested we include at least some lateral-view kinematic data, we have now included a new Figure 4 that not only illustrates this support triangle, but also how the joint angles have likely changed in Longshanks as a result of its increased length (see our response to Reviewer 3, below).

2) The general linear models should have included covariate-by-factor interactions.

> *Authors' response*: We re-ran our analyses using all three covariate-by-factor interaction terms. Out of all the GLM analyses (including the new analyses on the forelimb), only one revealed a significant interaction between body mass and line, in the timing of touch-down for the left hind paw in relation to the right forepaw. As argued by Engqvist (2005, *Animal Behaviour*, **70**, 967–971, in the absence of significant covariate-by-factor interaction terms, GLM analyses should be re-run without these terms, as failing to exclude them implies that the main effects of the factors cannot be generalized over the range of the covariate. Thus, we removed these terms and re-ran the analyses as before. In the case of the single significant interaction, we decided to remove this term as well, on the grounds that including it had only a modest effect on the response variable over the range of the covariate. We explain this process in detail in the updated Methods section (Lines 150-176).

3) Please clarify how stride length is calculated on the treadmill.

> *Authors' response*: the Treadscan program calculates stride length as the product of stride duration and belt speed. We have clarified this with a statement in the Methods section (Lines 136-137).

### Reviewer 2:

1) The article does not cite basic studies on the relationship between limb length and locomotion, and overlooks successful intra-population studies that illustrate the relationship between limb morphology and the metabolic cost of transport.

*Authors' response*: It was not our intention to suggest that there had not previously been studies that successfully demonstrated a relationship between limb morphology, locomotor biomechanics, and/or metabolic performance and selection *within* populations. In the original manuscript, we stressed that such studies had been "equivocal" rather than unsuccessful, and cited five studies in support of this claim, one of which found no relationship between limb morphology and metabolic cost (Minetti et al 2000), and four, including a key study suggested by the reviewer, by Steudel-Numbers and Tilkens (2004), that did show such a relationship exists.

In regards to Ted Garland's selection experiment, their selection protocol targets a behavioral trait (voluntary endurance running), not a morphological trait. To the best of our knowledge, the only study from Garland's group that tested for differences in limb morphology as a result of selection on wheel-running behavior (Garland and Freeman, 2005, *Evolution* **59** 1851-1854) found no difference in limb length between high-runners and random-bred controls.

We have now clarified our review of intraspecific/intra-population studies in the introduction. Specifically, we distinguish studies on the relationship between limb length and locomotor biomechanics, from those that address limb length and its effect on the metabolic cost of transport. We also stress that most intra-population studies that we could find have been done on humans, while studies in quadrupeds are much less common, though we have now included citations to a few relevant studies in dogs and horses (Lines 57-64, references 10-24).

2) There is no test of the impact of limb length on COT or on fitness, hence the experiment seems incomplete. Either the context should be narrowed, or the experiment should be integrated with physiology and fitness studies.

*Authors' response*: Unfortunately, we do not have proper physiological/metabolic, nor fitness data, collected from the same mice, to include in this study. Furthermore, we feel that such data should be published independently, and subsequently, to this one. With the inclusion of the new forelimb, gait sequence, and limb kinematic data (see below), we believe our key findings on gait differences in these mice stand on their own, and will be of interest to researchers who study the proximate relationships between limb morphology and locomotor biomechanics in an evolutionary context. Future studies on COT and fitness could then build on, and validate, the findings from this study. We have reframed the context of this study, to focus the aims and discussion on the proximate links between limb morphology and biomechanics in both limbs, while emphasizing that COT and fitness studies are natural follow-up studies that would help to fully "deconstruct" the complex links between morphology, performance, and evolutionary fitness.

3) Why were limbs not weighed, and inertial properties estimated, given the change in length relative to mass, and the variation in swing phases?

*Authors' response*: Most studies on limb inertial properties have been done on relatively large mammals such as primates. To the best of our knowledge, there have been no studies looking at inertial properties of the limbs in small animals such as mice, perhaps because of the difficulty in estimating these parameters at this scale. In this study, subtle differences in inertial properties in the groups would likely be smaller than measurement error. Moreover, the groups showed no difference in

swing phase durations (Table 3, Figure 2), which may be due to them having very similar inertial properties. We have now included a brief discussion of this question in Lines 271-277.

4) Why were limbs scanned rather than dissected and measured directly?

> *Authors' response*: As part of the Longshanks selection experiment, we are systematically collecting whole body uCT scans of the mice from each generation. The scans provide a more repeatable estimate of length across individuals, for two reasons. First, they require no dissection and "digitally" remove all soft tissues, such that skeletal measurements are not obstructed by connective and other soft tissues that remain after dissection, especially in the smaller bones of the hand and foot. Second, and related, the absence of soft tissues in the scans makes it easier to place digital landmarks precisely on the same anatomical features *across* individuals. We have added statements regarding the use and utility of uCT scans in the context of our experiment (Lines 178-199).

5) Why is there a discrepancy in the mean changes in stride length and frequency between the groups, since the two are related through belt speed?

> *Authors' response*: We thank the reviewer for bringing this discrepancy to our attention. We suspect it arose because we used the stride frequency value from TreadScan, which was based on the average of *all* stride cycles from its raw output, while we originally included only 3-4 steps for our analyses of swing/stance times and stride length. Moreover, TreadScan calculates stride frequency as the number of strides divided by the sum of the stride durations. We performed our updated analyses on the inverse of mean stride duration instead. Taking these potential sources of variation into account, our new values for stride length and stride frequency changes between the groups are more consistent with one another (both 7-8% mean increase in Longshanks, Table 3).

## Reviewer 3:

1) The authors only collected a bare minimum of locomotor gait variables, where a more complete suite was possible and should be encouraged. Specifically:

> a) Lateral view kinematic variables should be presented, even if they are limited

> > *Authors' response*: As stated in our original manuscript, we did not collect lateral view kinematic data from this sample of mice. Although we had low quality video (from a GoPro camera) for a few individuals, the off-center angles of these videos and poor lighting conditions prevented reliable data collection on joint locations and excursion angles. Instead, we obtained footage from a small sample of mice from a subsequent generation (F19) that were being filmed while running at the same speeds, for another project (n = 5 mice in each group). These videos allowed us to obtain approximate protraction, retraction and limb excursion angles for the hind limb from five steps in each mouse (Figure S1). We used these data to estimate the magnitude of limb angles, and to determine whether there were any significant differences between the lines in these angles (Table S1). Having found no difference between lines, we used pooled average retraction, protraction and limb excursion angle data, and step length in each line, to construct a support triangle for stance in Control and Longshanks, with known sides. This schematic diagram is shown in our new Figure 4. Using the known side lengths, which represent the functional length of the hind limb at touch-down and toe-off, and the

morphometric data for the hind limbs, we then estimated the changes in knee and ankle angles in Longshanks as a result of its modified hind limbs. These data preliminary show that limb joint angular changes in Longshanks are subtle, and thus unlikely to have impacted gait in Longshanks to a greater extent than the change in limb length itself. These new data are discussed in Lines 286-383.

       b) Forelimb gait data and between-limb accommodation/mismatch data should be presented

*Authors' response*: Our analyses now include the same gait variables (swing and stance durations, stride length and stride frequency) for the forelimb and hind limb. The forelimb data are included in their own GLM analysis (Tables 2 and 3, with speed and line as factors), as well as in a third GLM comparing the forelimb and hind limb between the limbs at 20 m/min (Table 4). They are presented graphically in our new Figure 2. The inclusion of the forelimb data shows that the differences in gait we uncovered in the hind limb are also present in the forelimb. The direct comparison of forelimb and hind limb gait variables (Table 4) also shows that, as expected, stride length, duration and frequency within lines are the same in both limbs, while swing times and stance times are slightly different in each limb. Thus, there appears to be no forelimb – hind limb mismatch or need for accommodation in the gait of Longshanks, most likely because, as our morphometric data indicate (Table 1), the forelimb also changed in length as a result of selection on the tibia.

       c) Gait sequence data should be presented

*Authors' response*: We now present data on the gait sequence in both lines running at both speeds. Specifically, using the initiation of stance in the right forepaw as a reference, we present the relative timing of touch-down for the other paws, expressed as a fraction of a full stride cycle (from 0 to 1). As with the other gait variables, we used a GLM analysis with speed and line as categorical factors, and the least squares means are presented in Table 5, and in our new Figure 3 (Lines 244-250, and 278-285). Figure 3 shows the gait sequence data for Longshanks superimposed on the sequence data for Controls at the faster speed only (there was no difference between the lines at either speed). As shown in Table 5 and Figure 3, the substantial increase in limb bone lengths in Longshanks did not appreciably change the gait sequence when compared to Control, presumably because there were similar changes in overall limb length in both fore- and hind limb. The relative duration of stance in the hind limb of Longshanks (i.e., its duty factor), is slightly increased at 20 m/min relative to Controls, though this difference is not significant (see dashed boxed in Figure 3).

We trust these substantive revisions to our manuscript have addressed the reviewers' concerns, and we thank them once again for their constructive suggestions. We have also made some minor edits to improve readability, and have addressed the formatting and other issues raised by the *PeerJ* editorial team. We look forward to your decision on our manuscript.

Best wishes,

Campbell Rolian

---

## Round 0.3 · accepted · Accept

· Academic Editor

Accept

Thank you for returning the manuscript promptly. It looks good to go.